# Leveraging pleiotropy for joint analysis of genome-wide association studies with per trait interpretations

**Kodi Taraszka**[1], **Noah Zaitlen**[2,3], **Eleazar Eskin**[1,3,4] *

**1** Department of Computer Science, University of California, Los Angeles, California, United States of America, **2** Department of Neurology, University of California, Los Angeles, California, United States of America, **3** Department of Computational Medicine, University of California, Los Angeles, California, United States of America, **4** Department of Human Genetics, University of California, Los Angeles, California, United States of America

* eeskin@cs.ucla.edu

**Data Availability Statement:** The data underlying the results presented in the study are available from the Neale lab: http://www.nealelab.is The method and simulation framework can be found: https://github.com/koditaraszka/pat.

## Abstract

We introduce pleiotropic association test (PAT) for joint analysis of multiple traits using genome-wide association study (GWAS) summary statistics. The method utilizes the decomposition of phenotypic covariation into genetic and environmental components to create a likelihood ratio test statistic for each genetic variant. Though PAT does not directly interpret which trait(s) drive the association, a per trait interpretation of the omnibus p-value is provided through an extension to the meta-analysis framework, m-values. In simulations, we show PAT controls the false positive rate, increases statistical power, and is robust to model misspecifications of genetic effect.

Additionally, simulations comparing PAT to three multi-trait methods, HIPO, MTAG, and ASSET, show PAT identified 15.3% more omnibus associations over the next best method. When these associations were interpreted on a per trait level using m-values, PAT had 37.5% more true per trait interpretations with a 0.92% false positive assignment rate. When analyzing four traits from the UK Biobank, PAT discovered 22,095 novel variants. Through the m-values interpretation framework, the number of per trait associations for two traits were almost tripled and were nearly doubled for another trait relative to the original single trait GWAS.

## Author summary

Genome-wide association studies have identified tens of thousands of genetic variants associated with complex traits. An ever increasing number of associated variants are shown to affect multiple traits, a phenomenon known as pleiotropy. We propose a method that leverages this genetic architecture and uses summary statistics to perform an omnibus association test between one genetic variant and a set of traits. Simulations show that the method properly controls for type-I errors and increases statistical power. In addition to a powerful omnibus test, we also enable a per trait interpretation of the associations by

**Funding:** K.T. and E.E. were supported by National Science Foundation grants 1910885 and 2106908, and National Institutes of Health grants K25-HL080079, U01-DA024417, P01-HL30568, P01-HL28481, R01-GM083198, R01-ES021801, R01-MH101782, and R01-ES022282. The funders had no role in study design, data collection and analysis, decision to publish, or preparation of the manuscript.

**Competing interests:** The authors have declared that no competing interests exist.

extending the m-value framework to account for the correlation structure between traits. This framework enables a significant increase in the identification of per trait effects.

## Introduction

Genome-wide association studies (GWAS) have been instrumental in identifying genetic variants associated with complex traits [1–3]. As a result, there are tens of thousands of unique associations in the GWAS catalog [4]. With ever increasing sample sizes in GWAS, more and more associated variants have been discovered. This suggests the presence of a large number of variants with small effect sizes that are not identified due to statistical power [5]. With the number of traits examined as well as sample sizes increasing over time, numerous variants are observed affecting more than one trait (i.e., pleiotropy) [6–10]. Some examples of pleiotropic effects include muscle mass and bone geometry, male pattern baldness and bone mineral density, as well as between multiple psychiatric disorders [11–13].

We hypothesize that because variants often affect more than one trait, we can leverage this pleiotropy to jointly analyze multiple traits. This would potentially increase statistical power and identify variants with even weaker effect sizes. Following this intuition, there have been many approaches for performing association tests using summary statistics across multiple traits [14–26]. While simultaneously analyzing multiple traits is advantageous for identifying novel variants, performing an omnibus test is inherently difficult to interpret. This is because an omnibus test assigns one p-value per variant for the set of traits, and it is not clear how to assign a per trait significance level in this context. Even when this is done, it is not straightforward to interpret due to issues such as inflation in false discovery rates when the assumption of homogeneity in effect sizes is violated [25].

In this paper, we propose an alternative framework with a two step procedure. First, all traits are jointly analyzed to produce one p-value for each variant. If this p-value is significant, it suggests that the variant is associated with one or more of the traits. To accomplish this first step, we develop an efficient method called pleiotropic association test (PAT) which leverages the estimated genome-wide genetic correlation between the traits to improve power and uses null simulations to accurately calibrate p-values. PAT also utilizes importance sampling to allow for estimation of significant p-values efficiently. The second step builds upon an interpretation framework first developed in the context of meta-analysis, m-values, to compute the posterior probability that a variant is associated with each trait [27]. We extend the m-value framework to take into account environmental and genetic correlation between traits.

In simulated data reflecting estimates of genetic and environmental covariance between real UK Biobank traits, we find that PAT is able to correctly control for false positives and increase power to identify novel associations. In comparisons to three multi-trait methods, MTAG, HIPO and ASSET, PAT has a 15.3% increase in the number of associations over the next best method [14, 24, 25, 28]. These results were then interpreted using the m-value framework where PAT identified 37.5% more per trait associations. Additionally while HIPO has only a 16.0% increase in power relative to MTAG for omnibus association testing, using the m-value framework to interpret HIPO's associations resulted in a 46.6% increase in per trait associations relative to MTAG. Finally, we analyzed four traits in the UK Biobank where PAT identified 22,095 novel variants and interpret the results for every trait using m-values. In two of the four traits, the number of per trait associations was almost three times greater than those found using the standard single trait GWAS, and it nearly doubled the number of per trait associations for another trait. PAT is freely available at https://github.com/koditaraszka/pat.

## Method

### Association testing in a single quantitative trait (GWAS)

We now describe the standard approach for determining if a genetic variant $g$ is associated with a quantitative trait $y$. Let $y$ and $g$ be measured for $N$ individuals where $g_j \in \{0, 1, 2\}$ is the minor allele count for each individual $j$. The column vector $g$ is then standardize according to the population proportion of the minor allele $p$ where $2p$ is the mean and $2p(1 - p)$ is the variance of $g$. This standardized column vector $x$ is defined as follows $x : x \in \left\{ \frac{-2p}{\sqrt{2p(1-p)}}, \frac{1-2p}{\sqrt{2p(1-p)}}, \frac{2-2p}{\sqrt{2p(1-p)}} \right\}$. The quantitative trait $y$ is normally distributed such that $y \sim \mathcal{N}(\mu, \sigma_e^2 I)$ where $\mu$ is the mean and $\sigma_e^2$ is the variance of the trait. This can then be mean-centered and scaled which results in the column vector $y \sim \mathcal{N}(0, I)$. We can now assume the following linear model:

$$y = \beta x + e$$

where $\beta$ is the effect size of the variant $x$ on the trait $y$ and the error $e$ follow the standard normal [2]. Ordinary Least Squares results in the estimator $\hat{\beta} = \frac{x^T y}{N}$ where $\hat{\beta} \sim \mathcal{N}\left(\beta, \frac{1}{N}\right)$. Setting $s = \frac{\hat{\beta}}{\hat{\sigma}_e} \sqrt{N}$ results in the following Gaussian: $s \sim \mathcal{N}\left(\frac{\hat{\beta}\sqrt{N}}{\hat{\sigma}_e}, 1\right)$.

We now test the null hypothesis: $x$ is *not* associated with $y$. More formally this tests if $\beta = 0$ or $s \sim \mathcal{N}(0, 1)$. The null model is rejected if $|s| > z$ where $z$ is the z-statistic at the $\alpha$ level of significance for the standard normal distribution. The corresponding critical value $z = \Phi^{-1}\left(1 - \frac{\alpha}{2}\right)$. Typically, human GWAS uses $\alpha = 5 \times 10^{-8}$ [3, 29, 30].

### Generalizing GWAS testing to multiple traits (MI GWAS)

As previously stated, GWAS traditionally analyzes each trait $y_i$ in a set of $T$ traits independently. In fact, each trait may be measured on distinct sets of individuals. Let us assume none of the traits $y_1, .., y_T$ have overlapping individuals; therefore every trait $y_i$ and the standardized genetic variant $x$ is measured for $N_i$ individuals. This assumption will later be relaxed. For now, the z-score for trait $i$: $s_i$ is tested for whether $|s_i| > z$, and this process is repeat for each trait independently. Another approach instead of performing $T$ different hypothesis tests is to determine whether the variant is associated with at least one of the traits. The corresponding null hypothesis is the variant is not associated with any of the traits. We refer to this method as multiple independent GWAS (MI GWAS). This results in $\beta_1 = \ldots = \beta_T = 0$ which is equivalent to saying the null model is $s_1 \sim \mathcal{N}(0, 1), \ldots, s_T \sim \mathcal{N}(0, 1)$. A simple way to test our null hypothesis is to check if the largest $s_i \in S = \{|s_1|, \ldots, |s_T|\}$ is greater than the critical value $z$ though $z$ will now need to be corrected for multiple testing. This can be done using a Bonferroni correction for the number of traits, $T$, so the critical value is $z = \Phi^{-1}\left(1 - \frac{\alpha}{2T}\right)$.

Another method for setting the critical value is using null simulations. This is done by simulating data according to $S = \{s_1, \ldots, s_T\}$ such that every $s_i$ is under the null hypothesis. As all traits are measured for different groups of individuals, there is no covariation between any pairs of traits. This means the multivariate $S \sim \mathcal{N}(0, \Sigma_e)$ has the identity matrix as its covariance matrix; therefore, we simulate $S \sim \mathcal{N}(0, I)$ $n$ times keeping the $\max\{|s_1|, \ldots, |s_T|\}$ for each $S$. We then sort the $n$ retained values and assign a p-value to each critical value using the quantile.

## Using pleiotropy for association testing in multiple traits (PAT)

Another method for hypothesis testing is a likelihood ratio test which compares the null model to a proposed alternative model. Currently, only the null model has been defined. For a single quantitative trait $y$ whose null hypothesis is $\beta = 0$ and $s \sim \mathcal{N}(0, 1)$, the alternative hypothesis is $\beta \neq 0$ and $\beta$ is assumed to follow a Gaussian distribution: $\beta \sim \mathcal{N}(0, \sigma_g^2)$, where $\sigma_g^2$ is the additive, per-variant heritabilty of the trait. As Gaussian distributions are conjugate priors to Gaussian likelihood functions, the distribution of $\beta$ can be used to get the Gaussian posterior predictive distribution, $s \sim \mathcal{N}(0, 1 + N\sigma_g^2)$. Two models that describe $s$ have been defined and result in the following likelihood ratio:

$$\frac{P(s|\mu = 0, \sigma^2 = 1 + N\sigma_g^2)}{P(s|\mu = 0, \sigma^2 = 1)} > \kappa$$

If the ratio of the likelihood functions is larger than $\kappa$, the null hypothesis is rejected. Before expounding on how to set $\kappa$, we will first extend the likelihood ratio test to the case of multiple traits. We retain the assumption that the traits are not measured on the same individuals. This means, there is no environmental correlation, so under the null hypothesis $S \sim \mathcal{N}(0, I)$. The assumption about distinct sets of individuals does not, however, have the same implication for the genetic correlation between genetic effects. Letting the $\mathrm{cov}(\beta_i, \beta_k) = \sigma_{g_{i,k}}$ we can derive $\mathrm{cov}(s_i, s_k) = \sqrt{N_i}\sqrt{N_k}\sigma_{g_{i,k}}$. This results in the alternative model being:

$$S \sim \mathcal{N}\left( \begin{pmatrix} 0 \\ \vdots \\ 0 \end{pmatrix}, \begin{pmatrix} 1 + N_1\sigma_{g_1}^2 & \cdots & \sqrt{N_1}\sqrt{N_T}\sigma_{g_{1,T}} \\ \vdots & \ddots & \vdots \\ \sqrt{N_1}\sqrt{N_T}\sigma_{g_{1,T}} & \cdots & 1 + N_T\sigma_{g_T}^2 \end{pmatrix} \right)$$

which can be written as $S \sim \mathcal{N}(0, \Sigma_e + \Sigma_g)$. This means under the alternative model, the covariance of $S$ is the sum of the environmental and genetic covariance where for now the environmental covariance is still the identity matrix, $I$. The likelihood ratio for PAT is now defined as:

$$\frac{P(S|\mu = 0, \Sigma = \Sigma_e + \Sigma_g))}{P(S|\mu = 0, \Sigma = \Sigma_e)} = \frac{P(S|\mu = 0, \Sigma = I + \Sigma_g))}{P(S|\mu = 0, \Sigma = I)} > \kappa$$

The critical value $\kappa$ is set for PAT using the same null simulations of $S \sim \mathcal{N}(0, I)$. This time the likelihood ratio for each $S$ is retained, sorted, and assigned a p-value using the quantile.

## Overlapping individuals for multiple traits

We now relax the assumption that no individual is measured for more than one trait. Under the null hypothesis, this means that $\Sigma_e$ in $S \sim \mathcal{N}(0, \Sigma_e)$ would not be the identity matrix $I$. In this case, $cov(s_i, s_k) = \frac{N_{shared}}{\sqrt{N_i}\sqrt{N_k}}\rho_{e_{i,k}}$. This means the covariance between $s_i$ and $s_k$ is the environmental correlation between the traits, and the environmental correlation is weighted by the proportion of overlapping individuals. Under the alternative hypothesis, we have $S : S \sim \mathcal{N}(0, \Sigma_e + \Sigma_g)$. We note that while sample overlap between traits affects $\Sigma_e$, it does not impact $\Sigma_g$.

## Importance sampling for null simulations

When performing null simulations, the number of simulations $n$ must be large enough that the critical value $\kappa$ is stable across estimates. In practice, this can require $n$ to be very large when $\alpha$ is really small because simulating $S : S \sim \mathcal{N}(0, \Sigma_e)$ with a likelihood ratio larger than $\kappa$ is expected to occur $\alpha \times n$ times. One method for reducing the number of simulations is importance sampling.

To explain our approach we first review importance sampling in one trait. While, traditionally $z = \Phi^{-1}\left(1 - \frac{\alpha}{2}\right)$ is used to set the critical value $z$ for the standard normal. It is also possible to use null simulations just as we do for MI GWAS and PAT. We simulate $s \sim \mathcal{N}(0, 1)$ $n$ times and sort $|s|$ and assign the p-values using the quantile. To obtain the significance of a specific critical value such as 5.2, enough null simulations must be performed to have a sufficient number of samples above the critical value. The p-value would then be estimated by counting the number of samples above the critical value divided by the total number of samples. Unfortunately, for very significant p-values this requires a very large number of samples since the vast majority of samples are below the critical value.

Importance sampling reduces the number of simulations needed for setting the critical value by simulating data according to a different distribution $v$ where $v$ results in samples larger than the critical value $z$ to occur more frequently. The procedure for estimating the p-value will then be adjusted to account for the differences between the two distributions, $s$ and $v$. In our approach, $v$ has the following distribution $v \sim \mathcal{N}(0, r1)$, and in Fig 1 the scaling factor $r = 2$ is used. In this figure, the critical value $z \approx 1.96$ for $\alpha = .05$ is shown for the null distribution $s$. We can see in Fig 1 that the distribution $v$ has many more samples in the tails; therefore, the significance level $\alpha$ does not correspond to $z$ for the distribution $v$. The p-value using importance sampling is estimated for each data point by first computing a weight $w$. This weight $w$ is the likelihood ratio $\frac{P(v|\mu=0,\sigma=1)}{P(v|\mu=0,\sigma=2)}$ of the data points from $v$ under the two models. By summing the weights of samples larger than the critical value and dividing by the sum of the weights for all samples, the p-value can be set for each critical value. We note that if $r = 1$, then $s$ and $v$ are identical and all the weights are 1. In this case, importance sampling and the standard approach are equivalent.

We can now extend this to learning about the null distribution of $S \sim \mathcal{N}(0, \Sigma_e)$ by simulating data according to the distribution $V : V \sim \mathcal{N}(0, r\Sigma_e)$. Again we will find that a well chosen alternative distribution $V$ results in more statistics greater than $\kappa$ in fewer simulations. The weight $w$ is the likelihood ratio $\frac{P(V|\mu=0,\Sigma=\Sigma_e)}{P(V|\mu=0,\Sigma=r\Sigma_e)}$ and will be used to obtain p-values as described above. When picking the scaling factor $r$, larger values will sample the tail in fewer simulations, however, care should be taken to ensure that a sufficient number of simulations are used to accurately set the critical threshold $\kappa$. In Table G in S1 Text, we found that $10^6$ consistently provided a stable estimate of the critical value for $\alpha = 5 \times 10^{-8}$ regardless of the choice of $r$.

## Interpreting GWAS meta-analyses

When performing an omnibus hypothesis test, there is only one p-value which cannot be directly interpreted on a per trait level. In previous work, the statistic m-values were introduced to enable interpretation of GWAS meta-analyses across studies, with m-values being the posterior probability of a genetic effect per study [27]. The original m-values assumed that across studies the effect sizes are similar as it considered the same trait across multiple studies. When applying this framework to multiple traits, the model needs to account for differing effect sizes to prevent spurious results. Below, we describe the extension to the m-value

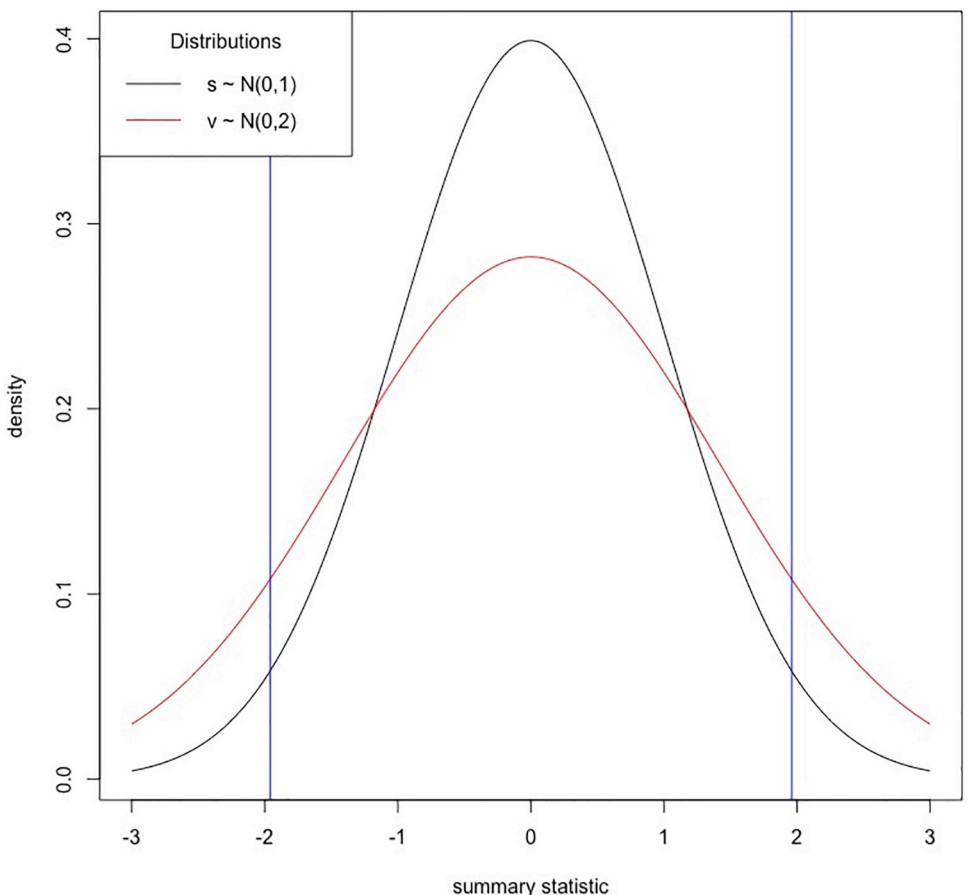

**Fig 1. Using importance sampling for setting critical values.** We simulated data according to two univariate Gaussian distributions $s \sim \mathcal{N}(0,1)$ and $v \sim \mathcal{N}(0,2)$ and show the densities. We show the critical value $z \approx 1.96$ for $\alpha = 0.05$. We would expect to see the critical value $|z|$ or larger more often when simulating data according to $v$ than when simulated under the distribution of $s$.

framework which assumes a random effects model for the genetic effect and that the effect sizes reflect the genome-wide estimate of genetic correlation.

We assume there are $T$ traits for which a variant has been identified as associated by PAT (or another omnibus test). While the variant is known to be associated, there are many possible configurations of an effect. There may be an effect in all traits in which case the configuration is $c = (1, \ldots, 1)$, or there may only be an effect in the first trait, $c = (1, 0, \ldots, 0)$. The set of all configurations can be written as $C = \{0, 1\}^T$ where $|C| = 2^T$. For each trait $i$, there is subset of configurations $C_i \subset C$ that are compatible with the variant having a genetic effect in that particular trait, where $|C_i| = 2^{T-1}$. This means that for every configuration $c \in C_i$, the $i$th index is always 1.

When PAT determines a variant is associated with the set of $T$ traits, it assumes the variant affecting all $T$ traits; therefore, the assumed posterior predictive distribution of effect is $P(S|\mu = 0, \Sigma = \Sigma_g + \Sigma_e)$. While pleiotropy is ubiquitous, the assumption that every variants affects all traits is not realistic. We will now define the genetic covariance matrix $\Sigma_g(c)$ that corresponds

to a configuration $c$ where

$$\Sigma_g(c) = \begin{cases} \sqrt{N_i}\sqrt{N_j}\sigma_{g_{i,j}}, & \text{if} \quad c_i = 1 \text{ and } c_j = 1 \\ 0, & \text{otherwise} \end{cases}$$

We note that when $c = (1, \ldots, 1)$, $\Sigma_g = \Sigma_g(c)$. With this in mind, m-values works by summing the posterior probabilities that corresponding to the configurations in $C_i$ and dividing by the the total sum of all posterior probabilities (set of configuration in $C$). Therefore, for each trait $i$:

$$m_i = \frac{\sum_{c \in C_i} P(S|\mu = 0, \Sigma = \Sigma_e + \Sigma_g(c))}{\sum_{c \in C} P(S|\mu = 0, \Sigma = \Sigma_e + \Sigma_g(c))}$$

where $S$ are summary statistics across the $T$ traits for one variant, and the m-value $m_i$ is the proportion of the all posterior probabilities compatible with there being an effect in trait $i$. When $m_i > 0.9$, the variant is assumed to be associated with the $i$th trait. Otherwise, the interpretation is left ambiguous.

While we assume the covariance structure of $\Sigma_g$ follows the polygenic model, for interpretation purposes this assumption is relaxed. Under the polygenic model, every variant has an effect; therefore, the expected effect size of each variant is $\frac{1}{M} \times h^2$ where $M$ is the total number of variants and $h^2$ is the estimated additive heritability of the trait. When only considering the variants found genome-wide significant, the expected effect size of these variants needs to be to estimated. We do this by estimating the number of causal variants $Q$ and rescale the genetic covariance matrix $\Sigma_g$ by $\frac{M}{Q}$ for the m-value interpretation framework.

This is necessary because $h^2 \in [0, 1]$ and with genome-wide association studies using millions of variants, $\Sigma_g + \Sigma_e \approx \Sigma_e$ under the polygenic model. While a valid model for association testing, distinguishing between different configurations of $\Sigma_g(c)$ to calculate the m-value is very difficult. Therefore, we scale $\Sigma_g$ and the resulting $\Sigma_g(c)$ by randomly selecting one associated variant per 100KB region for a total of $k$ variants. We then perform a grid search for $Q \in [1, M]$ and retain the value of Q which maximizes the likelihood function as shown below:

$$\underset{Q \in [1,M]}{\text{argmax}} \prod_{i=1}^{k} P\left(S_i|\mu = 0, \Sigma = \frac{M}{Q}\Sigma_g + \Sigma_e\right)$$

## Results

### Methods overview

**Pleiotropic association test.** Here we introduce PAT (pleiotropic association test) which takes in GWAS summary statistics measured for T traits and assumes each variant is drawn according to the multivariate normal (MVN) distribution: $S \sim \mathcal{N}(0, \Sigma)$. Furthermore, it assumes the covariance matrix can be decomposed into two independent components, environment and genetics ($\Sigma = \Sigma_e + \Sigma_g$). With this assumption in mind, PAT performs a likelihood ratio test (LRT) between two proposed MVN distributions. The null hypothesis is $\Sigma_g = 0$; therefore, the summary statistics for one variant, $S = \{s_1, \ldots, s_T\}$ has the following distribution: $S \sim \mathcal{N}(0, \Sigma_e)$.

Under the alternative hypothesis ($\Sigma_g \neq 0$), PAT models the genetic effect size according to the polygenic model and assumes the standard genetic correlation structure between traits

[31–33]. This results in summary statistics having the following distribution:
$S \sim \mathcal{N}(0, \Sigma_g + \Sigma_e)$.

Having now defined the distributions, a LRT can be computed for each variant's set of summary statistics $S$. Using the critical value $\kappa$ for the threshold of significance, it can now be decided whether a variant is associated with the set of traits.

$$\frac{P(S|\mu = 0, \Sigma = \Sigma_e + \Sigma_g)}{P(S|\mu = 0, \Sigma = \Sigma_e)} > \kappa$$

While likelihood ratio tests approximately follow a mixture of $\chi^2$ distributions, utilizing a $\chi^2$ distributions can be complicated and may have reduced power [34]. Therefore, instead of a closed form solution, PAT efficiently uses null simulations to determine significance (see Methods). Additionally, we note that when there is no environmental correlation ($\Sigma_e = I$), PAT is comparable to a Wald test.

**Multi-trait GWAS interpretation.** While PAT is a powerful tool for testing multi-trait associations, it is an omnibus test and only provides one p-value per variant. As a result, even when the null is rejected, we lack clarity as to which trait(s) drive the association; therefore, we propose a per trait p-value interpretation by estimating the posterior probability of a variant having a non-zero effect on a trait. This framework, m-values, was originally developed for interpreting meta-analysis across studies, but here it is extended to account for the covariance structure between traits [27].

To provide some intuition on m-values, we will describe the P-M plot (p-value by m-value plot) in Fig 2 [27]. This plot has the p-value from the original single trait GWAS along the y-axis and the corresponding m-value along the x-axis. A line at $-log(5 \times 10^{-8})$ denotes the threshold where a variant is considered genome-wide significant. Region A is where the original single trait GWAS resulted in the variant being significant while the interpretation of the omnibus test did not. There should not be data points in this region.

Regions B and D contain variants interpreted as associated with the trait because the m-value is greater than 0.9. Some of these variants have already been identified by the single trait GWAS (B) while other traits will be uniquely discovered on a per trait level (D). Region C contains the variants whose m-value is less than or equal to 0.9 and were left with an ambiguous interpretation.

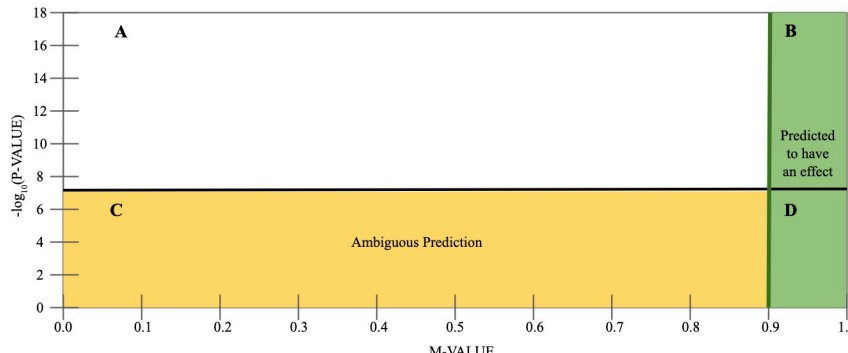

**Fig 2. Interpreting m-values using a P-M plot.** Along the x-axis is the per trait m-value and the y-axis shows the p-value from the original single trait GWAS. Region A is when the original association is significant, but the m-value interpretation is ambiguous. There should not be data points in this region. Region B and D are associations with an m-value greater than 0.9, so the interpretation is that there is a genetic effect in this trait. In Region C, the m-value interpretation is left ambiguous.

## Covariance structure between traits impacts the shape of PAT's rejection region

We now present an overview of PAT and its rejection region by comparing its shape to the rejection region of a version of standard GWAS generalized to multiple traits called multiple independent GWAS (MI GWAS). We chose to compare to this method over standard single trait GWAS because it accounts for multiple testing while being less stringent than a Bonferroni correction. MI GWAS works by testing if the largest summary statistic per trait was larger than the critical value for significance set using null simulations. For additional comparisons to MI GWAS and for comparisons of the rejection region of PAT to two additional methods: SUM and VC, see S1 Text [20].

In Fig 3, we simulated 100,000 summary statistics for two traits with the genetic variance ($\sigma_g^2 = 4.9 \times 10^{-5}$) and the sample size (N = 25,000) equal for both traits; the level of

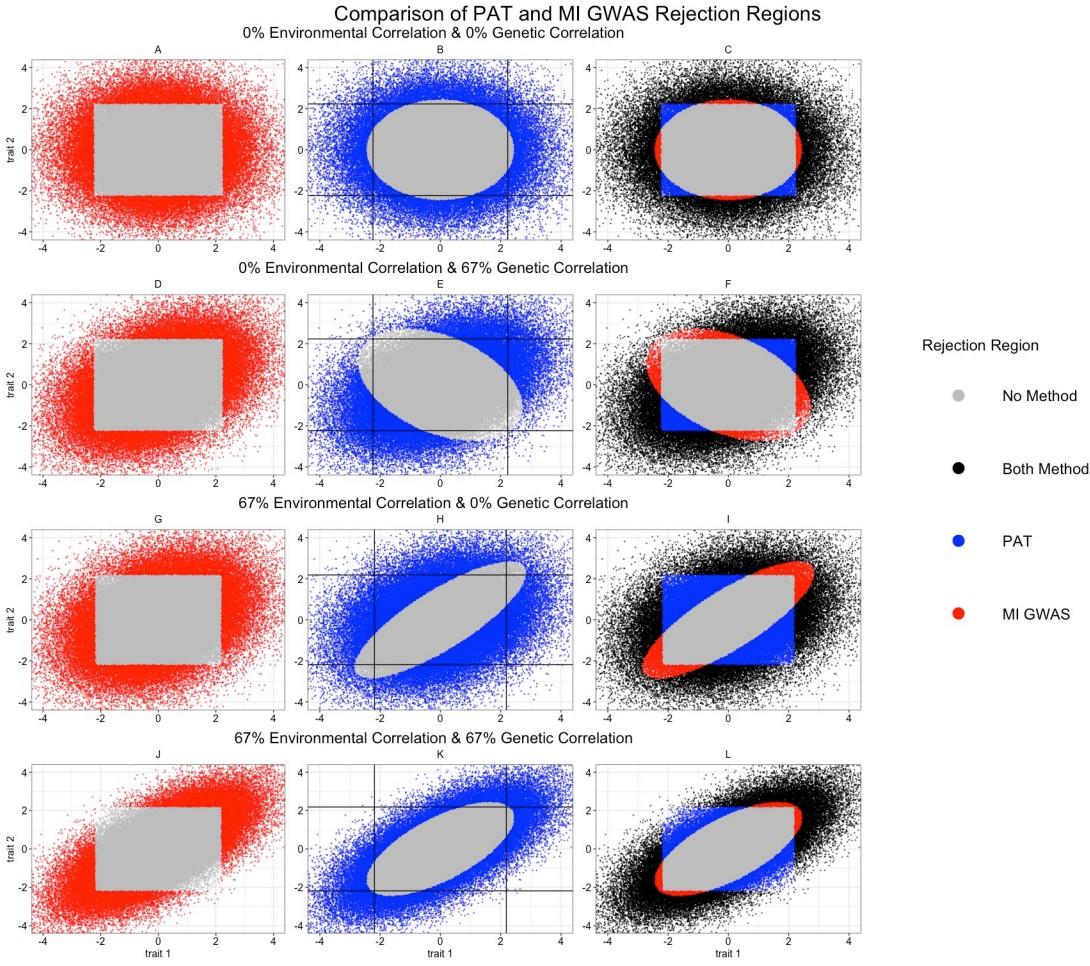

**Fig 3. Comparison of the rejection regions for MI GWAS and PAT.** We simulated 100,000 summary statistics for two traits with the genetic variance ($\sigma_g^2 = 4.9 \times 10^{-5}$) and the sample size (N = 25,000) equal for both traits. We varied the genetic and environmental correlation between traits and used $\alpha = 0.05$ for the level of significance. Each row corresponds to one set of simulations highlighting three points. The left column shows the rejection region of MI GWAS, the middle column has PAT's rejection region while the third column provides a comparison of the two methods. The simulations used in A-C have no environmental or genetic correlation while the data in D-F has no environmental correlation and 67% genetic correlation. For the third row of G-I, the environmental correlation was 67% while there was no genetic correlation between traits. The last row of simulations assumed an environmental and genetic correlation of 67%.

significance was $\alpha$ = 0.05. The first column highlights MI GWAS's performance. Variants which were correctly identified as associated are shown in red while the ones missed by MI GWAS are grey. In each row regardless of model specification, the shape of MI GWAS's rejection region was a square. As MI GWAS does not account for genetic correlation, there was no effect on the critical value when this parameter varied.

The same phenomenon was not true for PAT which is shown in the second column (with the critical values of MI GWAS depicted with black lines). Here, when PAT rejected the null hypothesis, the data points are in blue, and those PAT failed to reject are grey. In all four rows the shape of the rejection region was an ellipse. As PAT models environmental and genetic correlation, both parameters impacted the shape of the elliptical rejection region. In the first row there is no environmental or genetic correlation, so the shape was exactly a circle. This means any extreme value for at least one of the summary statistics was likely to be rejected. In the second row, we modeled 67% genetic correlation and no environmental correlation. Here, the shape enabled PAT to correctly identify more variants with positively correlated z-scores but failed to aid in identifying variants with negatively correlated z-scores. This follows the intuition that modeling genetic correlation would increase power to identify variants whose summary statistics followed this correlation pattern.

While the first two rows followed intuition, the shape of PAT's rejection region in the last two rows was less intuitive. In the third row, we simulated traits with 67% environmental correlation but no genetic correlation. In this situation, the shape of the rejection region was in the direction of the environmental correlation; therefore, PAT has more power when z-scores were negatively correlated relative to when they were positively correlated. This means when summary statistics were positively correlated, PAT failed to reject the null unless the values were very extreme because it assumed the only source of positive correlation was the environment. The gain in power in the direction of negative correlation was due to the same idea that these values were unlikely under a positively correlated environment unless there was a non-environmental effect (i.e., genetics). In the final row, we simulated a positively correlated environment and genetics. Here, the shape of the rejection region still followed the direction of the environmental correlation. This aided in controlling false positives, but it meant that PAT may have been overly conservative in the direction of environmental correlation even when there was genetic correlation in the same direction. Further to that point, the critical value for MI GWAS as shown in Fig 3H and 3K (black lines) was identical. In Fig 3K, there were fewer variants pass this cut-off that were missed by PAT relative to Fig 3H. This means that while PAT was consistently more conservative in the direction of the environmental correlation, it was less conservative when it expected a genetic reason for correlated summary statistics.

The right most column has a comparison of the relative power of PAT and MI GWAS. Variants that were correctly identified by both methods are black data points while those missed by both are grey. The variants only identified as significant by PAT are blue while those found only by MI GWAS are red. Under all four simulation frameworks, PAT had more statistical power than MI GWAS with the greatest improvement occurring when there was genetic correlation and no environmental correlation (Fig 3F). We note that the blue region may appear smaller in Fig 3F than in Fig 3I and 3L, but the density of the data points was higher (see Table H in S1 Text).

## M-values provide accurate interpretation of omnibus association tests

In the previous section, we provided an overview of how PAT provides a per variant omnibus p-value for the set of traits. Here, we used simulated data reflective of four real UK Biobank traits (see Tables A-D in S1 Text) to provide some intuition about m-values as well as highlight

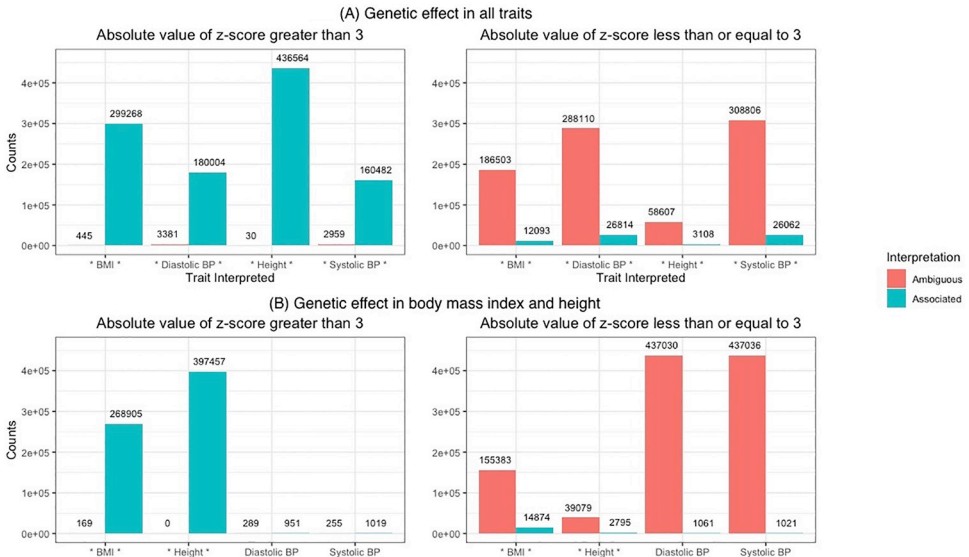

**Fig 4. Interpreting per trait associations from omnibus significant variants.** We simulated one million variants for four traits under two models. The first set of simulations assumed there was a genetic effect in every trait (A), while the second model only has a genetic effect in body mass index and height (B). The associated traits are noted with an asterisks (*). The results for each trait were split based on the absolute value of the z-score and showed the interpretation as either ambiguous or associated. The threshold for associated is an m-value greater than 0.9.

its accuracy. M-values were produced by enumerating over the set of configurations $C = \{0, 1\}^4$ which indicate which trait(s) have a genetic effect, $\Sigma_g(c)$. We note that the configuration $c = (1, 1, 1, 1)$ indicates a genetic effect in all four traits (i.e. $\Sigma_g(c) = \Sigma_g$). For each configuration, we calculated the posterior probability $P(S|\mu = 0, \Sigma = \Sigma_e + \Sigma_{g(c)})$. We then take the sum of the configurations compatible with trait $i$ ($c_i = 1$) and divide by the total probability over all configurations to produce the m-value for trait $i$. If this ratio $m_i > 0.9$, we interpreted the omnibus variant-trait association to be an association between the variant and trait $i$. If this ratio $m_i \leq 0.9$, we left the interpretation as ambiguous. We note that m-values are a Bayesian quantity whose threshold is a matter of convention established in previous work [27]; in Fig B in S1 Text, we gave some empirical support for this threshold as the m-value can be seen as a very conservative bound on the false assignment rate.

In Fig 4, we simulated one million variants under two model conditions. In Fig 4A, there was a genetic effect in all four traits: body mass index, diastolic blood pressure, height, and systolic blood pressure while in Fig 4B, we modeled a genetic effect in only body mass index and height. In Fig 4, the truly associated traits were denoted with an asterisks (*) around the trait name. The effect sizes were simulated such that the first model has 50% power and 44% when there was a genetic effect in only body mass index and height. We split the summary statistics (z-scores) for each trait based on whether there was even modest signal in a particular trait (|z-score| > 3). This distinction was due to differing expectations on the ability to correctly interpret an association. We note that the inclusion of variants with a |z-score| ≤ 3 for a particular trait was primarily done for completeness and their interpretation was overwhelmingly ambiguous (Fig 4 right panel). We therefore, focus on the left panel of Fig 4 where the |z-score| > 3.

When there was a genetic effect in all four traits (top row left side), the m-value was greater than 0.9 for the vast majority of z-scores which means the majority of variants were correctly interpreted as associated with all traits. Diastolic and systolic blood pressure had the most ambiguous associated variants with 3,381 and 2,959, respectively. This, however, was still less

than 2% of the variants with at least a modest effect size ($|z\text{-score}| > 3$) being interpreted as ambiguous for each trait. Furthermore, when there was a modest effect size the overall false negative rate was 0.6% across the traits.

The second set of simulations modeled a genetic effect in only two of the traits (bottom row left side). Here, the m-value framework correctly interpreted when there was a genetic effect in body mass index and height for most significant variants. For body mass index only 169 of the variants were missed and none were left ambiguous for height. For diastolic blood pressure and systolic blood pressure, approximately 1,200 variants for each trait that had a $|z\text{-score}| > 3$. The m-value wrongly identified 951 and 1,019 of those variants as associated for diastolic blood pressure and systolic blood pressure, respectively. Overall, 99.5% of the variants analyzed for diastolic blood pressure and systolic blood pressure were left with an ambiguous interpretation. For body mass index and height, 64.5% and 90.5% of all variants, respectively, were correctly interpreted as associated when there was only a genetic effect in these two traits.

These simulations show the m-value framework has a low false positive assignment rate, and enabled the correct classification of many associated variants. This was especially true when $|z\text{-score}| > 3$ while $|z\text{-score}| \leq 3$ typically resulted in the interpretation being ambiguous regardless of the ground truth. While this was still a false negative assignment, many of these associations would have failed to pass a nominal test for significance (p-value $<0.05$).

## PAT is a powerful method for omnibus association testing in multi-trait GWAS

Now that we have established the intuition behind PAT, it is important to understand its performance relative to other multi-trait methods. Here, we compare four methods: PAT, MTAG, HIPO, and ASSET [14, 24, 25]. HIPO is an omnibus method that performs eigenvalue decomposition resulting in orthogonal components each of which is used to create a weighted sum of z-scores. For this comparison, z-scores from all components are considered simultaneously and a variant is deemed associated as long as it is genome-wide significant for at least one component. Another method is MTAG, and it also uses a weighted sum of z-scores. MTAG, however, is not an omnibus method but tests each trait separately while leveraging information from the other traits. The results from MTAG are converted to an omnibus test by determining if the variant is genome-wide significant for at least one trait. The final method is ASSET; this method works by searching for the subset of traits with the strongest positive signal and separately the strongest negative signal. ASSET then combines these test statistics using a chi-squared method to form an overall test statistic which we use for comparison. While MTAG and HIPO generate multiple test statistics for each variant, we do not correct for multiple testing; all methods are tested at $\alpha = 5 \times 10^{-8}$.

In Table 1, 1.5 million z-scores were simulated for four traits with the environmental and genetic covariance structure based on traits from the UK Biobank and 10% (150,000) of the variants were causal in at least one trait. The first row in Table 1 corresponds to the 1,350,000 variants simulated under the null. All four methods correctly identified zero associated variants. The remaining 150,000 truly associated variants were equally split across all configurations of genetic effect. For each of the configurations, there were three scaling factors for the heritability covariance matrix, $\Sigma_{h2}$, which can be thought of as the number of causal variants ($\frac{\Sigma_{h2}}{\text{causal}}$) where causal equals 40k, 24k, or 16k. We note that the methods assume the polygenic model (i.e. $\Sigma_g = \frac{\Sigma_{h2}}{\#\,\text{variants}}$), but we simulated assuming fewer causal variants to create effect sizes large enough that there was power for discovery. The 10,000 variants for each configuration were split such that 5k, 3k, and 2k simulations came from each of the respective causal effect sizes. The configurations are subsets of the four traits, body mass index ($\mathcal{B}$), diastolic blood

**Table 1. Comparison of multi-trait GWAS methods.** 1.5 million variants were simulated with z-scores for four traits with 10% of variants as truly associated. The first column lists which trait has a genetic effect. The second column is the number of variants simulated under this specific model. The third column is the genetic effect size. The remaining four columns contain the number of variants identified as associated by four methods: PAT, HIPO, MTAG, and ASSET. The final row of the table contains each methods running time.

| Genetic Effect | Number of Variants | $\frac{\Sigma_{h2}}{\text{causal}}$ | Genome-Wide Significant | | | |
|---|---|---|---|---|---|---|
| | | | PAT | HIPO | MTAG | ASSET |
| No Trait | 1,350,000 | 0 | 0 | 0 | 0 | 0 |
| $\mathcal{B},\mathcal{D},\mathcal{H},\mathcal{S}$ | 5,000 | 40,000 | **113** | 54 | 60 | 103 |
| | 3,000 | 24,000 | **198** | 113 | 119 | 196 |
| | 2,000 | 16,000 | 291 | 194 | 196 | **326** |
| $\mathcal{B},\mathcal{D},\mathcal{H}$ | 5,000 | 40,000 | **108** | 53 | 56 | 76 |
| | 3,000 | 24,000 | **204** | 113 | 121 | 170 |
| | 2,000 | 16,000 | **307** | 216 | 226 | 286 |
| $\mathcal{B},\mathcal{D},\mathcal{S}$ | 5,000 | 40,000 | 0 | 2 | **3** | 1 |
| | 3,000 | 24,000 | 0 | **12** | 7 | 5 |
| | 2,000 | 16,000 | 0 | **26** | 12 | 11 |
| $\mathcal{B},\mathcal{H},\mathcal{S}$ | 5,000 | 40,000 | **124** | 73 | 58 | 92 |
| | 3,000 | 24,000 | **216** | 166 | 128 | 199 |
| | 2,000 | 16,000 | **352** | 281 | 219 | 334 |
| $\mathcal{D},\mathcal{H},\mathcal{S}$ | 5,000 | 40,000 | **88** | 28 | 36 | 56 |
| | 3,000 | 24,000 | **161** | 105 | 111 | 131 |
| | 2,000 | 16,000 | **257** | 173 | 176 | 227 |
| $\mathcal{B},\mathcal{D}$ | 5,000 | 40,000 | 0 | **2** | 1 | 0 |
| | 3,000 | 24,000 | 0 | **15** | 6 | 2 |
| | 2,000 | 16,000 | 0 | **34** | 4 | 1 |
| $\mathcal{B},\mathcal{H}$ | 5,000 | 40,000 | **96** | 36 | 40 | 60 |
| | 3,000 | 24,000 | **160** | 106 | 116 | 138 |
| | 2,000 | 16,000 | **260** | 196 | 201 | 255 |
| $\mathcal{B},\mathcal{S}$ | 5,000 | 40,000 | 0 | **33** | 11 | 5 |
| | 3,000 | 24,000 | 5 | **81** | 32 | 30 |
| | 2,000 | 16,000 | 12 | **128** | 61 | 48 |
| $\mathcal{D},\mathcal{H}$ | 5,000 | 40,000 | **90** | 40 | 42 | 41 |
| | 3,000 | 24,000 | **177** | 111 | 114 | 127 |
| | 2,000 | 16,000 | **253** | 195 | 185 | 204 |
| $\mathcal{D},\mathcal{S}$ | 5,000 | 40,000 | 0 | **3** | 0 | 0 |
| | 3,000 | 24,000 | 0 | **7** | 2 | 2 |
| | 2,000 | 16,000 | 0 | **23** | 14 | 9 |
| $\mathcal{H},\mathcal{S}$ | 5,000 | 40,000 | **80** | 40 | 32 | 46 |
| | 3,000 | 24,000 | **185** | 127 | 94 | 131 |
| | 2,000 | 16,000 | **225** | 191 | 144 | 179 |
| $\mathcal{B}$ | 5,000 | 40,000 | 0 | **12** | 8 | 4 |
| | 3,000 | 24,000 | 1 | **32** | 20 | 9 |
| | 2,000 | 16,000 | 6 | **51** | 45 | 30 |
| $\mathcal{D}$ | 5,000 | 40,000 | 0 | **5** | 1 | 0 |
| | 3,000 | 24,000 | 0 | **14** | 4 | 1 |
| | 2,000 | 16,000 | 1 | **35** | 15 | 7 |
| $\mathcal{H}$ | 5,000 | 40,000 | **89** | 36 | 46 | 47 |
| | 3,000 | 24,000 | **154** | 82 | 92 | 94 |
| | 2,000 | 16,000 | **191** | 126 | 139 | 134 |

(*Continued*)

**Table 1.** (Continued)

| Genetic Effect | Number of Variants | $\frac{\Sigma_{h2}}{\text{causal}}$ | Genome-Wide Significant | | | |
|---|---|---|---|---|---|---|
| | | | **PAT** | **HIPO** | **MTAG** | **ASSET** |
| $\mathcal{S}$ | 5,000 | 40,000 | 0 | **11** | 1 | 0 |
| | 3,000 | 24,000 | 0 | **39** | 3 | 2 |
| | 2,000 | 16,000 | 1 | **66** | 4 | 2 |
| Total | 1,500,000 | — | **4,405** | 3,486 | 3,005 | 3,820 |
| Running Time (seconds) | — | — | 72 | 96 | 150 | 54,709 |

pressure ($\mathcal{D}$), height ($\mathcal{H}$), and systolic blood pressure ($\mathcal{S}$). The final row in Table 1 contains the running time for each method. Here, we see ASSET was significantly slower than the other three methods which were comparable to each other. PAT's efficient running time indicates that the use of importance sampling can enables a speed up comparable to deriving p-values analytically; the differences in compute time between PAT, HIPO, and MTAG were likely due to other factors (e.g. MTAG does a number of sanity checks prior to analysis).

While no simulation framework truly reflects the real world, this arrangement attempted to non-exhaustively model different scenarios that occur when analyzing z-scores from multiple traits. Namely, we explored the power to discover summary statistics with different causal effect sizes and violations of a pleiotropic effect in all traits. Under the various configurations shown here, all of the methods were under powered due to the simulations being centered around zero; however, PAT was the most powerful method in nearly half of the simulated scenarios as well as overall. Across all scenarios PAT identified 4,405 associated variants which was an 15.3% increase over ASSET (3,820), a 26.4% increase over HIPO (3,486) and a 46.6% increase over MTAG (3,005). While PAT generally performed the best, the other methods did significantly better when the genetic effect in height was absent. Without considering environmental correlation, this scenario was similar to that seen in Fig 3B and 3E. There we saw that the closer one trait's z-score was to 0, the larger the other trait's effect size needed to be. Another factor was the environmental correlation; the other three traits have more environmental correlation to each other than to height which was similar to the scenario in Fig 3K. In this case, PAT was shown to be conservative in the direction of environmental correlation. To better understand the effect of environmental correlation on statistical power, we conducted further simulations in the supplementary materials (see Fig A and Table E in S1 Text). Finally, we explore the simulations from this section on a per trait level below.

## M-values enable more per trait interpretations in multi-trait GWAS

The four multi-trait methods were previously compared in regards to their power to perform omnibus association testing (see above). Here, we investigated the per trait interpretation of these associations. As MTAG computes a p-value for every trait, the method provides a direct per trait interpretation; therefore, for each respective trait we reported the variants with a p-value $<5 \times 10^{-8}$. The method, ASSET, considers all possible subsets and selects the one that maximizes its test statistic. This is done separately in the positive and negative directions of effect and are then combined for a two-tailed test which determines the omnibus association. For the associated variants, we tested each direction separately for significance (p-value $<5 \times 10^{-8}$ and interpreted the subsets that produced a significant association as the trait(s) driving the association. The last two methods, HIPO and PAT, only provided an omnibus interpretation; therefore, we applied the m-value framework to assign a per trait association to variants whose omnibus p-value $<5 \times 10^{-8}$. For both methods, this was done by taking the

associated variants and calculating the posterior predictive probability (m-value) of whether there was a genetic effect in each particular trait. If the m-value was greater than 0.9, the variant was deemed associated with the trait. Otherwise, the interpretation was left ambiguous.

Prior to exploring the per trait interpretation, we note that only MTAG controls the false positive per trait interpretation due to its use of p-values for the assignment; m-values do not directly control for false positives. As a result, m-values are only meant to provide empirical insights and interpretation to p-values not replace them. This means the comparisons in Table 2 between MTAG's p-values and the m-value interpretations are not an apples to apples comparison. In Fig C in S1 Text, we provide a fairer comparison by ranking the p-values and m-values. There we show that for any false positive rate, PAT and HIPO have more true positive per trait assignments than MTAG. Separately, we acknowledge that while ASSET provides the subset of traits with the strongest association signal with the intent of a more interpretable multi-trait association. It is possible that a trait was included in the optimal subset due to its tagging the causal signal in another trait. In this case, including the trait was useful for increasing the association power but would lead to an incorrect interpretation.

In Table 2, all methods analyzed 1.5 millions simulations with 10% (150,000) causal variants equally divided across all configurations of genetic effect. For each of the configurations, different effect sizes were also considered (see above). In Table 2, the number of per trait associations was reported by trait under each configuration. When the variant did not truly have a genetic effect on the trait, the box was greyed to indicate false positives. Overall, Table 2 resembled the results shown in Table 1.

One example of an exception was when there was a genetic effect in body mass index, height, and systolic blood pressure $(\mathcal{B}, \mathcal{H}, \mathcal{S})$. While PAT identified more associated variants, HIPO has more per trait associations for systolic blood pressure. This means that while HIPO has less power than PAT for the omnibus test (see Table 1), it was able to provide the most per trait interpretations for this trait. This was due to HIPO identifying different associated variants than PAT which were then interpreted on a per trait level. We also saw this phenomenon when there was a genetic effect in height and systolic blood pressure $(\mathcal{H}, \mathcal{S})$.

Overall, PAT identified 6,264 true per trait associations from its 4,405 omnibus associations. For HIPO, the m-value framework interprets 4,557 true per trait associations from its 3,486 significant variants. When comparing PAT to HIPO, there are 37.5% more true per trait associations than HIPO due to PAT having more power as an omnibus method. The method, ASSET, identified 3,820 significant associations with 3,944 traits correctly placed in the optimal subset. Finally, we consider MTAG which directly identified 3,064 total per trait associations (3,005 omnibus associations). While HIPO and PAT identified 16.0% and 46.6% more omnibus associations than MTAG, respectively the m-value framework enabled a 48.7% increase for HIPO and a 104.4% increase for PAT in per trait associations relative to MTAG, a method designed for per trait interpretation. When comparing HIPO and PAT and their m-values to ASSET, we saw that HIPO had 8.7% fewer omnibus associations than ASSET but 15.5% more per trait assignments. Separately, while PAT had 15.3% more omnibus associations than ASSET, there were 58.8% more per trait findings.

While m-values enabled a significant increase in per trait interpretations, as stated before, the m-value threshold does not directly control for false positives. In Table 2, MTAG had no false positive per trait associations. The m-values produced for PAT and HIPO, however, did result in a small number of false positive assignments, 58 and 42 respectively. This was 0.92% and 0.91% of their respective per trait interpretations. When we considered the subsets produced by ASSET, we observed there were 28 false positive placements (0.73%).

**Table 2. Four multi-trait GWAS methods with per trait interpretation.** 1.5 million variants were simulated with z-scores for four traits with 10% of variants being truly associated. The first column lists which trait has a genetic effect. The second column is the number of variants simulated under this specific model. The third column is the genetic effect size of the variant. The remaining columns are split by trait where the performance of the four methods are shown for each trait. These 16 columns present the number of variants identified as associated by each method for the specific trait. MTAG uses p-values, ASSET uses the optimal subset, while PAT and HIPO use the m-value framework to provide per trait associations.

| Genetic Effect | Number of Variants | $\frac{\Sigma_{h2}}{causal}$ | Body Mass Index | | | | Diastolic Blood Pressure | | | | Height | | | | Systolic Blood Pressure | | | |
|---|---|---|---|---|---|---|---|---|---|---|---|---|---|---|---|---|---|---|
| | | | PAT | HIPO | MTAG | ASSET | PAT | HIPO | MTAG | ASSET | PAT | HIPO | MTAG | ASSET | PAT | HIPO | MTAG | ASSET |
| $\mathcal{B,D,H,S}$ | 5,000 | 40,000 | **28** | 17 | 10 | 15 | **5** | 3 | 0 | 4 | **113** | 44 | 50 | 74 | **8** | 2 | 0 | 1 |
| | 3,000 | 24,000 | **64** | 38 | 17 | 39 | **31** | 15 | 2 | 10 | **198** | 96 | 98 | 143 | **35** | 22 | 3 | 16 |
| | 2,000 | 16,000 | **131** | 89 | 44 | 78 | **71** | 44 | 8 | 21 | **284** | 157 | 139 | 221 | **70** | 56 | 18 | 47 |
| $\mathcal{B,D,H}$ | 5,000 | 40,000 | **34** | 15 | 8 | 12 | **11** | 10 | 1 | 4 | **108** | 43 | 47 | 56 | 1 | 1 | 0 | 1 |
| | 3,000 | 24,000 | **64** | 42 | 26 | 43 | **41** | 22 | 7 | 16 | **200** | 87 | 89 | 118 | 2 | 1 | 0 | 1 |
| | 2,000 | 16,000 | **139** | 96 | 47 | 85 | **85** | 61 | 18 | 29 | **301** | 182 | 171 | 224 | 2 | 3 | 0 | 0 |
| $\mathcal{B,D,S}$ | 5,000 | 40,000 | 0 | 0 | 0 | 0 | 0 | **1** | 1 | 0 | 0 | 0 | 0 | 0 | 0 | **2** | 2 | 1 |
| | 3,000 | 24,000 | 0 | 0 | 0 | 0 | 0 | **8** | 5 | 3 | 0 | 0 | 0 | 0 | 0 | **10** | 3 | 5 |
| | 2,000 | 16,000 | 0 | 0 | 0 | 0 | 0 | **15** | 7 | 9 | 0 | 0 | 0 | 0 | 0 | **21** | 5 | 6 |
| $\mathcal{B,H,S}$ | 5,000 | 40,000 | **39** | 29 | 12 | 12 | 3 | 2 | 0 | 1 | **124** | 52 | 46 | 64 | 16 | **25** | 0 | 1 |
| | 3,000 | 24,000 | **74** | 60 | 25 | 49 | 3 | 1 | 0 | 0 | **215** | 108 | 101 | 145 | 51 | **63** | 3 | 17 |
| | 2,000 | 16,000 | **160** | 125 | 53 | 92 | 4 | 3 | 0 | 0 | **345** | 206 | 170 | 244 | 97 | **115** | 6 | 40 |
| $\mathcal{D,H,S}$ | 5,000 | 40,000 | 1 | 1 | 0 | 0 | **6** | 1 | 1 | 4 | **88** | 26 | 35 | 46 | **8** | 3 | 0 | 6 |
| | 3,000 | 24,000 | 1 | 1 | 0 | 0 | **24** | 19 | 3 | 6 | **161** | 98 | 107 | 116 | **25** | 18 | 3 | 6 |
| | 2,000 | 16,000 | 3 | 1 | 0 | 1 | **53** | 37 | 9 | 11 | **257** | 162 | 163 | 195 | **61** | 40 | 8 | 21 |
| $\mathcal{B,D}$ | 5,000 | 40,000 | 0 | 0 | 0 | 0 | 0 | **1** | 1 | 0 | 0 | 0 | 0 | 0 | 0 | 1 | 0 | 0 |
| | 3,000 | 24,000 | 0 | 0 | 0 | 0 | 0 | **14** | 6 | 2 | 0 | 0 | 0 | 0 | 0 | 0 | 0 | 0 |
| | 2,000 | 16,000 | 0 | 0 | 0 | 0 | 0 | **32** | 4 | 1 | 0 | 1 | 0 | 0 | 0 | 0 | 0 | 0 |
| $\mathcal{B,H}$ | 5,000 | 40,000 | **26** | 14 | 6 | 15 | 2 | 0 | 0 | 1 | **96** | 31 | 34 | 50 | 1 | 0 | 0 | 1 |
| | 3,000 | 24,000 | **57** | 44 | 28 | 29 | 0 | 0 | 0 | 1 | **159** | 88 | 90 | 110 | 2 | 1 | 0 | 1 |
| | 2,000 | 16,000 | **116** | 102 | 64 | 76 | 0 | 0 | 0 | 0 | **255** | 151 | 144 | 177 | 1 | 1 | 0 | 0 |
| $\mathcal{B,S}$ | 5,000 | 40,000 | 0 | **19** | 11 | 5 | 0 | 1 | 0 | 0 | 0 | 0 | 0 | 0 | 0 | **18** | 0 | 1 |
| | 3,000 | 24,000 | 5 | **47** | 30 | 26 | 0 | 0 | 0 | 0 | 0 | 1 | 0 | 1 | 3 | **56** | 2 | 3 |
| | 2,000 | 16,000 | 10 | **85** | 54 | 41 | 0 | 0 | 0 | 0 | 0 | 0 | 0 | 2 | 6 | **74** | 7 | 11 |
| $\mathcal{D,H}$ | 5,000 | 40,000 | 1 | 0 | 0 | 0 | 7 | 5 | 0 | 0 | **90** | 37 | 42 | 40 | 0 | 1 | 0 | 0 |
| | 3,000 | 24,000 | 3 | 0 | 0 | 2 | 36 | 23 | 3 | 4 | **177** | 100 | 111 | 117 | 0 | 0 | 0 | 0 |
| | 2,000 | 16,000 | 3 | 2 | 0 | 3 | 87 | 81 | 18 | 24 | **253** | 174 | 173 | 185 | 3 | 2 | 0 | 0 |
| $\mathcal{D,S}$ | 5,000 | 40,000 | 0 | 0 | 0 | 0 | 0 | **1** | 0 | 0 | 0 | 0 | 0 | 0 | 0 | **2** | 0 | 0 |
| | 3,000 | 24,000 | 0 | 0 | 0 | 0 | 0 | **4** | 1 | 2 | 0 | 0 | 0 | 1 | 0 | **4** | 2 | 2 |
| | 2,000 | 16,000 | 0 | 0 | 0 | 0 | 0 | **14** | 8 | 7 | 0 | 0 | 0 | 0 | 0 | **17** | 6 | 5 |
| $\mathcal{H,S}$ | 5,000 | 40,000 | 1 | 0 | 0 | 0 | 1 | 2 | 0 | 0 | **80** | 33 | 32 | 40 | **12** | 10 | 0 | 3 |
| | 3,000 | 24,000 | 1 | 0 | 0 | 1 | 2 | 1 | 0 | 1 | **185** | 99 | 92 | 119 | 39 | **48** | 2 | 11 |
| | 2,000 | 16,000 | 5 | 2 | 0 | 6 | 2 | 1 | 0 | 0 | **225** | 155 | 141 | 166 | 54 | **83** | 3 | 17 |
| $\mathcal{B}$ | 5,000 | 40,000 | 0 | **12** | 8 | 4 | 0 | 0 | 0 | 0 | 0 | 0 | 0 | 0 | 0 | 0 | 0 | 0 |
| | 3,000 | 24,000 | 1 | **32** | 20 | 9 | 0 | 0 | 0 | 0 | 0 | 0 | 0 | 0 | 0 | 0 | 0 | 0 |
| | 2,000 | 16,000 | 6 | **51** | 45 | 30 | 0 | 0 | 0 | 0 | 0 | 0 | 0 | 1 | 0 | 1 | 0 | 0 |
| $\mathcal{D}$ | 5,000 | 40,000 | 0 | 1 | 0 | 0 | 0 | **4** | 1 | 0 | 0 | 0 | 0 | 0 | 0 | 0 | 0 | 0 |
| | 3,000 | 24,000 | 0 | 0 | 0 | 0 | 0 | **14** | 4 | 1 | 0 | 0 | 0 | 0 | 0 | 0 | 0 | 0 |
| | 2,000 | 16,000 | 0 | 0 | 0 | 0 | 1 | **33** | 15 | 6 | 0 | 0 | 0 | 0 | 0 | 1 | 0 | 0 |
| $\mathcal{H}$ | 5,000 | 40,000 | 1 | 1 | 0 | 0 | 1 | 1 | 0 | 0 | **89** | 36 | 46 | 47 | 1 | 1 | 0 | 0 |
| | 3,000 | 24,000 | 2 | 1 | 0 | 1 | 1 | 1 | 0 | 0 | **154** | 82 | 92 | 94 | 1 | 1 | 0 | 1 |
| | 2,000 | 16,000 | 0 | 0 | 0 | 0 | 2 | 1 | 0 | 0 | **191** | 126 | 139 | 133 | 1 | 1 | 0 | 0 |

(*Continued*)

**Table 2.** (Continued)

| Genetic Effect | Number of Variants | $\frac{\Sigma_{h2}}{causal}$ | Body Mass Index | | | | Diastolic Blood Pressure | | | | Height | | | | Systolic Blood Pressure | | | |
|---|---|---|---|---|---|---|---|---|---|---|---|---|---|---|---|---|---|---|
| | | | PAT | HIPO | MTAG | ASSET | PAT | HIPO | MTAG | ASSET | PAT | HIPO | MTAG | ASSET | PAT | HIPO | MTAG | ASSET |
| $\mathcal{S}$ | 5,000 | 40,000 | 0 | 0 | 0 | 0 | 0 | 0 | 0 | 0 | 0 | 0 | 0 | 0 | 0 | **11** | 1 | 0 |
| | 3,000 | 24,000 | 0 | 0 | 0 | 0 | 0 | 0 | 0 | 0 | 0 | 0 | 0 | 0 | 0 | **39** | 3 | 2 |
| | 2,000 | 16,000 | 0 | 0 | 0 | 0 | 0 | 0 | 0 | 0 | 0 | 0 | 0 | 0 | 1 | **66** | 4 | 2 |

## PAT discovers novel per trait associations in the UK Biobank

While simulations have indicated PAT is a powerful method for association testing and m-values enable a per trait interpretation, we now apply this two step approach to real data. We analyzed the UK Biobank summary statistic for body mass index, diastolic blood pressure, height, and systolic blood pressure [28]. Here, five methods were compared: Single Trait GWAS (how the z-scores and p-values were derived), MTAG, MI GWAS, HIPO and PAT [14, 25]. The set of variants were processed such that only variants which were biallelic, have non-ambiguous strands, a minor allele frequency greater than 1%, and an INFO score greater than 80% were retained. This left 7,025,734 variants that meet the criteria for all four traits. The reference and alternate allele were coordinated across traits by flipping the direction of the effect when necessary. LD-Score regression and cross-trait LD-Score regression were used to calculate the genetic and environmental covariance structure (see S1 Text) [35, 36].

Using standard single trait GWAS, there were 211,546 uniquely associated variants across the four traits of interest. With MTAG, 164,263 uniquely associated variants were identified, 931 of which were novel associations. MI GWAS implicated 183,669 variants as associated, but none of the variants were novel discoveries due to MI GWAS having less power than single trait GWAS by design. When analyzing the traits with HIPO, 177,519 associated variants were found with 19,829 being new variants. PAT identified 200,112 uniquely associated variants with 22,095 being novel. None of the multi-trait methods identified more distinct variants than the standard single trait GWAS though MTAG, HIPO, and PAT identified new variants. This was likely due to insufficient power to capture variants associated with only one trait. For further exploration of the omnibus results see the Table F in S1 Text.

When comparing the methods on their per trait associations, more associations were identified by leveraging multiple traits. While standard single trait GWAS identified 211,546 uniquely associated variants, only 18,764 were implicated as associated with more than one trait for a total of 233,540 associations as reported in Table 3. When analyzing the traits using MTAG, 18,054 out of 164,263 uniquely associated variants were found to be associated with

**Table 3. UK Biobank data interpretation.** We analyzed four traits from the UK Biobank using five methods: Single Trait GWAS, MTAG, MI GWAS, HIPO, and PAT and show the variants associated with each trait. For Single Trait GWAS and MTAG, the per trait association was directly computed. For MI GWAS, HIPO and PAT, an omnibus association was first performed. The significant variants were then interpreted using the m-value framework using 0.9 as the threshold.

| Trait Interpreted | Directly From GWAS | | M-value >0.90 | | |
|---|---|---|---|---|---|
| | Single Trait GWAS | MTAG | MI GWAS | HIPO | PAT |
| body mass index ($\mathcal{B}$) | 37,205 | 32,527 | 64,706 | 65,462 | **67,139** |
| diastolic blood pressure ($\mathcal{D}$) | 18,593 | 17,610 | 56,369 | **58,294** | 56,271 |
| height ($\mathcal{H}$) | 160,227 | 117,882 | 155,730 | 136,519 | **191,420** |
| systolic blood pressure ($\mathcal{S}$) | 17,515 | 16,927 | 48,308 | **51,234** | 49,125 |
| Total | 233,540 | 184,946 | 325,113 | 311,509 | **363,955** |

more than one trait. This resulted in there being a total of 184,946 per trait associations. While single trait GWAS and MTAG provided a per trait p-value, MI GWAS, HIPO, and PAT did not. In order to interpret their associations, a per trait m-value must be assigned. When using the m-value framework, MI GWAS interpreted 325,113 per trait associations due to 96,519 of its 183,669 associated variants being associated with more than one trait. Out of the set of 183,669 uniquely associated variants, there were 8,213 whose interpretation was left ambiguous. This means that while those variants were significantly associated with the set of traits according to the omnibus test, the interpretation as to which of the traits was still ambiguous. HIPO identified 177,519 associated variants where 94,5333 were interpreted as associated with more than one trait. There were 862 with an ambiguous interpretation while 311,509 were interpreted as associated with at least one trait. Finally, the m-value framework was applied to PAT resulting in 363,955 per trait associations from the set of 200,112 unique variants. Out of which 111,126 variants were interpreted as associated with more than one trait and 9,869 were left with an ambiguous interpretation.

We note that while MI GWAS cannot by definition have more power than Single Trait GWAS, once a variant was implicated as associated with at least one trait the interpretation could be assigned to multiple traits. This means that as long as the effect size in one trait was large enough to result in MI GWAS finding the variant significant, the weaker effect sizes could still be interpreted using m-values. This is because the m-value framework leveraged the genetic and environmental covariation between traits regardless of whether or not the original method modeled it which enables an increase in per trait associations. In fact, PAT had over 100,000 more per trait associations than single trait GWAS in Table 3 even though it implicated fewer variants. For body mass index, PAT, MI GWAS and HIPO almost doubled the number of per trait associations and nearly tripled it for systolic blood pressure. For diastolic blood pressure, the number of per trait associations was more than tripled due to the m-value framework. In Table 3, MTAG performed on par with single trait GWAS on a per trait level. One reason for the difference in performance was the nature of the methods. For MI GWAS, HIPO, and PAT, the variant was first implicated and then interpreted on a per trait basis while MTAG and single trait GWAS assigned statistical significance for each trait separately.

## Replicating novel associations in the GIANT consortium

The three methods PAT, HIPO, and MTAG respectively identified 22,095, 19,829, and 931 novel associations when jointly analyzing four traits from the UK Biobank. For PAT, all novel associations had an m-value greater than 0.9 in at least one trait which means all associations had a per trait interpretation. The breakdown of the per trait associations were: 12,261 variants interpreted as associated with body mass index, 7,868 with diastolic blood pressure, 21,119 with height, and 7,605 were interpreted as associated with systolic blood pressure. For HIPO, there were 862 associations with an ambiguous interpretation. The breakdown of the 18,967 variants with a per trait interpretation were: 6,202 associated with body mass index, 8,420 with diastolic blood pressure, 6,396 with height, and 9,844 with systolic blood pressure. For MTAG which provided per trait p-values, 33 of the 931 novel associations were associated with body mass index, 254 with diastolic blood pressure, zero with height, and 644 were associated with systolic blood pressure. Now equipped with novel per trait associations, these discoveries should be validated in an external dataset; therefore, we used the GIANT consortium to see if any of the new associations for body mass index or height could be reproduced [37, 38].

For body mass index, the European summary statistics from the GIANT consortium contained 2,554,638 variants which were separately matched to the variants identified by PAT, HIPO, and MTAG using the RSID, reference, and alternate allele and had a minor allele

frequency reported. When the reference and alternate allele differed between the two data sets, the direction of the effect size in the replication data set was flipped. After identifying which variants were present in both data sets, the variants discovered by PAT and HIPO were clumped by taking the largest m-value (i.e., posterior predictive probability) and removing all other variants within a 1MB region. We clumped variants on the m-value instead of the p-value due to the p-value's significance potentially being driven by a different trait. For MTAG, as there was a per trait p-value, we clumped variants using the minimum p-value. Out of the 12,261 novel variants discovered by PAT and interpreted to be associated with body mass index in the UK Biobank, 3,946 were found in the GIANT consortium which resulted in 408 independent variants after clumping. Separately, for the 6,202 variants identified by HIPO, 2,111 were found in the GIANT consortium which resulted in 218 independent variants after clumping. Of the 33 novel variants discovered by MTAG, ten were found in the GIANT consortium which resulted in six independent variants after clumping.

This process was repeated in height where the GIANT consortium had 2,550,859 variants to be considered. Out of the 21,119 variants identified by PAT and interpreted as associated with height in the UK Biobank, there are 7,068 also found in the GIANT data set. After clumping these variants to the peak m-value per megabase region, there were 735 independent associations. Separately, for the 6,396 variants identified by HIPO, 2,216 were also in the GIANT consortium which resulted in 196 independent variants after clumping. MTAG identified zero novel variants associated with height.

In order to test the replication rate, we performed a one-sided z-test in the direction of the effect size ($\beta$) in the UK Biobank. Beginning with PAT, we saw that for body mass index, 378 out of 408 variants (92.6%) had their effect sizes in the same direction in both cohorts. We tested each variant for replication using the level of significance $\alpha = \frac{0.05}{408} = 1.22 \times 10^{-4}$ and found 14 variants replicated. For height, 97.4% (716) of the tested variants had their effect sizes in the same direction. For replication, we set the level of significance to $\alpha = \frac{0.05}{735} = 6.80 \times 10^{-5}$. Here, we saw that 197 of the 735 variants replicated. Separately, we considered the variants discovered by HIPO and saw that for body mass index, 209 of 218 (95.9%) had their effect sizes in the same direction in both cohorts and 27 had a p-value below $\alpha = \frac{0.05}{218} = 2.29 \times 10^{-4}$. For height, 189 of the 196 (96.4%) independent variants discovered by HIPO and interpreted as associated with height had effect sizes in the same direction in both cohorts and 43 had a p-value below $\alpha = \frac{0.05}{196} = 2.55 \times 10^{-4}$. For MTAG, the level of significance for body mass index was $\alpha = \frac{0.05}{6} = 8.33 \times 10^{-3}$. We observed two variants with a p-value below this threshold; all six variants (100%) had their effect sizes in the same direction in both cohorts.

For the variants that failed to replicate, there were a number of possible reasons this occurred. In Table 4, we explored how statistical power affected our replication rate. We note that MTAG was not included in the table and that all six variants had a replication power over 90% with a mean of 97.2%. When considering the variants discovered by PAT and HIPO, we first binned the variants into deciles by their replication power. For each decile, we calculated the average power to replicate the effect sizes observed in the UK Biobank in the GIANT consortium. We note that the GIANT consortium did not release the minor allele frequency observed in their samples but instead provided the minor allele frequency observed in Hap-Map [29]. While a reasonable estimate, inaccuracies in the minor allele frequency impact power calculations. Additionally, we note that the GIANT consortium summary statistics were from a meta-analysis which may have a lower effective sample size than the reported sample size due to heterogeneity between cohorts. For PAT, the average power over all variants for body mass index was 39.4% while it was 64.1% for HIPO, however we only saw a replication rate of 3.4% and 12.4%, respectively. For height, the replication rate for PAT was 26.8% and

**Table 4. Replication power in the GIANT consortium for BMI and height.** We tested the novel associations in the UK Biobank discovered by PAT and HIPO for replication in the GIANT consortium. We separately clumped using the lead variant as determined by the m-value. For each variant, we calculate replication power and bin the variants into deciles. The first column lists the trait. The second column is the decile while the third and fourth column are the average power within the set for each respective method. The number of variants tested for replication, the expected number of replications, and the number of variants that replicated are reported in the next six columns. The final two columns contain the number of variants with effect sizes from the GIANT consortium in the same direction seen in the UK Biobank. A binomial test on whether the proportion of effect sizes in the same direction across studies is greater than 50% of all tested variants in the set. A single asterisks means the results are significant at the nominal $\alpha = 0.05$ and two asterisks indicates significance at $\alpha = \frac{0.05}{20}$.

| Trait | Power | Average Power | | Number of Variants Tested | | Expected Number of Replications | | Number of Replications | | Number of Effect Sizes in Same Direction | |
|---|---|---|---|---|---|---|---|---|---|---|---|
| | | PAT | HIPO | PAT | HIPO | PAT | HIPO | PAT | HIPO | PAT | HIPO |
| Body Mass Index | 0–10% | 6.8% | 8.5% | 44 | 5 | 3 | 0 | 0 | 0 | 35** | 4 |
| | 10–20% | 15.2% | 14.2% | 92 | 24 | 14 | 3 | 0 | 0 | 84** | 23** |
| | 20–30% | 24.7% | 25.3% | 57 | 7 | 14 | 2 | 1 | 0 | 54** | 7* |
| | 30–40% | 35.5% | 34.2% | 40 | 10 | 14 | 3 | 0 | 1 | 37** | 9* |
| | 40–50% | 45.3% | 46.1% | 33 | 12 | 15 | 6 | 2 | 1 | 32** | 12** |
| | 50–60% | 55.5% | 55.3% | 23 | 9 | 13 | 5 | 0 | 1 | 23** | 9** |
| | 60–70% | 64.5% | 65.4% | 43 | 18 | 28 | 12 | 5 | 1 | 40** | 18** |
| | 70–80% | 75.2% | 76.6% | 53 | 66 | 40 | 51 | 1 | 10 | 51** | 64** |
| | 80–90% | 85.9% | 82.9% | 15 | 45 | 13 | 37 | 1 | 8 | 15** | 44** |
| | 90–100% | 92.6% | 93.6% | 8 | 22 | 7 | 21 | 4 | 5 | 7* | 19** |
| Height | 0–10% | 4.2% | 4.9% | 15 | 7 | 1 | 0 | 1 | 0 | 12* | 7* |
| | 10–20% | 13.4% | 15.6% | 11 | 30 | 1 | 5 | 2 | 4 | 11** | 27** |
| | 20–30% | 24.7% | 24.4% | 13 | 25 | 3 | 6 | 4 | 2 | 12** | 24** |
| | 30–40% | 36.0% | 34.6% | 30 | 28 | 11 | 10 | 5 | 3 | 26** | 27** |
| | 40–50% | 45.7% | 44.4% | 46 | 18 | 21 | 8 | 5 | 2 | 46** | 17** |
| | 50–60% | 55.5% | 55.2% | 90 | 16 | 50 | 9 | 27 | 0 | 89** | 16** |
| | 60–70% | 65.8% | 64.6% | 135 | 20 | 89 | 13 | 38 | 5 | 130** | 20** |
| | 70–80% | 75.0% | 75.2% | 303 | 21 | 227 | 16 | 92 | 10 | 300** | 21** |
| | 80–90% | 83.1% | 83.4% | 75 | 26 | 62 | 22 | 21 | 14 | 73** | 25** |
| | 90–100% | 93.0% | 91.7% | 17 | 5 | 16 | 5 | 2 | 3 | 17** | 5* |

was 21.9% for HIPO, but the overall power was 65.5% and 47.3%, respectively. In Table 4, we observe for both traits that as power increased so did the replication rate; however, neither trait replicated at the expected rate. The only exception was in height when the power was between 0–30%, we saw PAT replicating slightly over the expected rate.

While we have shown that our replication rate was below expectation, the expected replication rate was likely overestimated due to winner's curse [39]. As the variants tested for replication were not identified as associated by the original single trait GWAS, these variants have sub-optimal power for discovery. This means these variants have small effect sizes and were only found associated after leveraging their covariance structure with other traits. As a result, the bias in the effect size will be much larger here than in variants that were already well powered for discovery. For the non-replicating variants, further power increases (e.g. larger sample sizes) are essential to better tease out which variants warrant follow up analyses.

While many variants failed to replicate potentially due to insufficient power or winner's curse, we also tested whether the effect sizes were in the same direction between GIANT and the UK Biobank. If the variant truly had no effect on the trait, the concordance of effect size across the data sets should be 50%; however, if there was a genetic effect, a higher concordance across data sets is expected. We performed a binomial test in each decile to determine whether the proportion of effect sizes in the same direction was greater than 0.50. Using the

significance threshold $\alpha$ = 0.05, every test was significant for PAT and 19/20 were significant for HIPO. As there were 20 tests, we adjusted for multiple testing using a Bonferroni correction ($\alpha = \frac{0.05}{20}$). All but two tests were statistically significant at this new threshold for PAT and 15/20 were significant for HIPO. A test of overall concordance in body mass index tested whether $\frac{378}{408} = 0.926$ was greater than 0.5 returned a p-value of $4.34 \times 10^{-78}$. We also tested height ($\frac{716}{735} = 0.974$) which returned a p-value of $1.06 \times 10^{-184}$. Separately, for HIPO, we tested the proportion of variants in the same direction in body mass index ($\frac{209}{2018} = 0.959$; $p = 6.43 \times 10^{-51}$) and in height ($\frac{189}{196} = 0.964$; $p = 2.04 \times 10^{-47}$). Therefore, we conclude that while the actual replication rate was low, there is evidence of real genetic signal in the variants identified by the multi-trait methods.

## Discussion

Here, we presented PAT, a method that leveraged pleiotropy for joint association testing in multiple traits as well as an extension to the m-value framework. Through simulations, PAT was shown to control the false positive rate as well as significantly increase statistical power to detect pleiotropic effects. The impact of misspecifying model parameters on PAT was also explored. We saw that PAT was robust to there being a genetic effect in some subsets of traits while other configurations significantly impacted PAT's performance. One major limitation of PAT was its lack of per trait interpretations. This was overcome by the extension to the m-value framework presented here. M-values enabled a per trait interpretation of PAT and other omnibus methods. Through simulations, we found that the false positive assignment rate from m-values was low.

Additionally, PAT was compared to three multi-trait methods: MTAG, HIPO, and ASSET. While PAT was shown to be a more powerful method for omnibus association testing, there were some scenarios where PAT was underpowered. One such scenario was when there was high environmental correlation. In this scenario, HIPO and MTAG provided better models for joint analysis of traits due to PAT's conservative nature in the presence of strong environmental correlation.

While we primarily considered how the methods handled the misspecification of the covariance structure between traits. Another fundamental difference between PAT and the other methods was how PAT derived its critical value using null simulations. In contrast, the other methods produced their p-values analytically; however, they could also leverage null simulations. This may be particularly beneficial to HIPO whose signal was likely spread across multiple components. Accounting for this empirically may better calibrate the global null. Further work would need to be done to explore this, but we note that importance sampling as used here would enable an efficient solution.

In addition to simulations, PAT analyzed four traits in the UK Biobank and discovered 22,095 novel associations while the next best method, HIPO, identified 19,829. After computing m-values and clumping the per trait associations, the replication of associated variants in body mass index and height were tested in the GIANT consortium. For body mass index, 14 of 408 independent variants discovered by PAT replicated while 27 of 218 independent variants replicated for HIPO. The replication rate in height was much higher with 197 out of 735 variants replicating for PAT and 43 of 196 for HIPO. While the replication rate was below expectation, this may be due to winner's curse [39]. The variants tested for replication were novel discoveries that were under powered in the original association. In addition to testing replication, we tested whether the effect sizes between the UK Biobank and GIANT consortium were in the same direction. Overall, there was significant evidence that the direction of the effect sizes were concordant which is improbable under the null.

While PAT was shown to be an effective method for leveraging pleiotropy between traits, the optimal number of traits to jointly model was not explored. As the number of traits increase, the genome-wide estimate of genetic correlation ceases to hold across all traits. This would result in fewer novel associations as the power gains would be stunted by model misspecification. Further exploration is needed to determine which traits should be analyzed together and how to effectively cluster the traits into these sets.

Another limitation to PAT was it assumed the genetic covariance structure was constant across the genome. PAT was agnostic to the environmental and genetic covariance between traits and treated these as input. As all variants were tested independently, the user could input a different covariance structure for each variant or a set of variants. This may enable a significant power increase as modeling local covariance structure better reflects the covariance structure between z-scores [40–42]. Our method, however, only considered the global estimate of genetic and environmental correlation between traits and further work is needed to quantify the impact of such modifications on both power and false positives which other's have explored [19].

One limitation to the m-value interpretation framework was how it estimated the number of causal variants for the genetic covariance matrix. Currently, for association testing the genetic covariance matrix was scaled according to the polygenic model (i.e. all variants were causal). Once variants were implicated as associated, we used grid search to find the genetic covariance matrix scaling that best reflected the average effect size of independent variants. This in effect was an approximation to the number of causal variants. Further work is merited to better estimate the number of causal variants for each trait as well as the number shared between traits.

## Supporting information

**S1 Text. Additional simulations and a table of the real data results.**
(PDF)

## Author Contributions

**Conceptualization:** Kodi Taraszka, Eleazar Eskin.

**Formal analysis:** Kodi Taraszka, Noah Zaitlen, Eleazar Eskin.

**Investigation:** Kodi Taraszka, Noah Zaitlen, Eleazar Eskin.

**Methodology:** Kodi Taraszka, Eleazar Eskin.

**Software:** Kodi Taraszka, Eleazar Eskin.

**Supervision:** Noah Zaitlen, Eleazar Eskin.

**Visualization:** Kodi Taraszka, Noah Zaitlen, Eleazar Eskin.

**Writing – original draft:** Kodi Taraszka, Noah Zaitlen, Eleazar Eskin.

**Writing – review & editing:** Kodi Taraszka, Noah Zaitlen, Eleazar Eskin.

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
