## [Decision Letter · Decision Letter 0]

14 Apr 2022

Dear Dr Eskin,

Thank you very much for submitting your Research Article entitled 'Leveraging pleiotropy for joint analysis of genome-wide association studies with per trait interpretations' to PLOS Genetics.

The manuscript was fully evaluated at the editorial level and by independent peer reviewers. The reviewers appreciated the attention to an important problem, but raised some substantial concerns about the current manuscript. Based on the reviews, we will not be able to accept this version of the manuscript, but we would be willing to review a much-revised version. We cannot, of course, promise publication at that time.

If you decide to revise the manuscript for further consideration at PLOS Genetics, please aim to resubmit within the next 60 days, unless it will take extra time to address the concerns of the reviewers, in which case we would appreciate an expected resubmission date by email to plosgenetics@plos.org.

[LINK]

We are sorry that we cannot be more positive about your manuscript at this stage. Please do not hesitate to contact us if you have any concerns or questions.

Yours sincerely,

Michael P. Epstein

Associate Editor

PLOS Genetics

David Balding

Section Editor: Methods

PLOS Genetics

Reviewer's **Comments to the Authors:**

Reviewer #1: ## Overview

This work introduces a new likelihood-based approach, PAT, for GWAS summary statistic data to test the global null hypothesis of no association across multiple traits. PAT uses estimates of the genetic covariance from cross-trait LD-score regression to calculate a likelihood ratio test statistic where the p-values are estimated using a computationally efficient approach. The authors find that PAT makes overall more discoveries compared to MTAG/HIPO on simulated and real data. In addition to PAT, the authors calculate posterior probabilities for each trait to help determine which traits may be driving statistical significance by extending a previous quantity called m-value.

## Major comments

- Can the authors make the software/reproducible code for the analysis/simulations available?

- There should be a set of simulations exploring the impact of the environmental correlation among traits since PAT’s improvements seem to stem from low environmental correlation (or is it because height has a negative genetic correlation with the other traits?). In either case, I think this should be explored a bit more in the paper. For example, varying the proportion of traits with no environmental correlation from 0, 1/4, 1/2, 3/4, 1 and the magnitude of the correlation (none, low, moderate, high). I think this is important to include as a motivator for PAT and the prevalence of such analysis in practice. It could also be worth exploring the direction of the genetic correlations as that may be important in the method comparisons.

- If I interpreted it correctly then the authors are determining statistical significance in HIPO if any component passes the threshold. However, with eigenvectors, the signal can get spread out across multiple components. I suspect this is happening in the simulations/real data because height is negatively correlated with the other traits. Can the authors combine the p-values across these components and then generate an empirical null similar to PAT to estimate empirical p-values. An initial thought could be to combine via Fisher's method and then probit-transform this p-value into a z-value. You can then estimate the empirical null for this statistic in a similar way as PAT. This is probably a better comparison to test the global null that leverages all the information across the components.

- Shouldn't the estimated number of causal SNPs be different for each entry in the genetic covariance for the m-value optimization?

- Does the calculation of m-values have to take into account the selection process (i.e., only calculate m-values when p < alpha)? So the actual distribution in the equation is not Gaussian but instead a truncated Gaussian?

## Minor comments

- Can the authors also include the computational time for each method in Table 1?

- m-values are a simple and effective method to help determine important traits. The discussion in the main text about m-values/z-values and the corresponding threshold of 0.9 was a bit confusing. m-values are a Bayesian quantity and the threshold is somewhat arbitrary in practice? The quantity itself is just a measure of the strength of evidence? If so, I think it should be made a bit clearer (and potentially simplified) in the main text.

- It would be useful to add a couple sentences in the methods section about choosing r for the importance sampling algorithm (and then reference the section in the supplement).

## Typos

- Double check the equations: there are small math mode typos across document (e.g., "max", "causal" not in text mode while in math environment)

- Page 16: In the regression model, e is the random error not the residual.

- Figure 2: Might want “s_1”, “s_2” to be in math mode. Also, might want to reorder the labels on figure legend so “No method” is on the bottom/top.

- Page 7/33: Supplementary Fig 3. -> Fig in main text

- Page 2: “Fiallly,” -> “Finally”

Reviewer #2: The authors presented a powerful new method for multi-trait genetic association analysis, and a modified m-value for per-trait interpretation. In general, I found the method interesting, and the paper clearly written. However, I have the following comments:

Table 2 compared the number of per-trait associations discovered by PAT and HIPO using m-values and MTAG using p-values. What is the p-value threshold for MTAG? Since p-values are not directly comparable to m-values, the authors need to justify the choice of p-value that is comparable to m-value>0.9. I would also suggest using precision-recall curve to compare the three methods, which is free of the effects of threshold choice.

I found the third column of Figure 2 a bit confusing for comparing MI GWAS and PAT, mainly because the density of dots is not well reflected and hence performance is not proportional by the area of each color. I suggest adding the proportion of dots falling into each color. It may also be helpful to use a smoothed density plot with color gradient proportional to density.

Table 4 and page 14 shows low replication rate in the GIANT consortium. I would suggest including the replication rate of HIPO and MTAG to get a sense of the relative performance.

The paper mainly uses the number of significantly associated variants as a metric of performance. This is okay in simulations, but in real data analysis (e.g. Table 3) it would be more interpretable to use the number of independent genetic loci.

In Tables 1 and 2, the definition of \\sigma_g/causal is a bit confusing. If they refer to genetic effect size, why are they such large numbers? More description would be appreciated.

Minor comments:

Page 2 fifth line from bottom: “finally” instead of “fially”.

Page 16 line 7 of section Generalizing GWAS testing to multiple traits (MI GWAS): “instead of” duplicated.

Reviewer #3: The authors describe and address a pressing and common problem in genetic epidemiology: how to identify variants that (a) show evidence for pleiotropy (i.e. are associated with two or more traits) and (b) how to identify which traits a potentially pleiotropic variant is associated with. As the authors nicely review, there are many methods that leverage data on multiple traits to increase power to test the global null hypothesis that a variant is not associated with any of the studied traits--but these are not tests of pleiotropy per se (where the null is that the variant is associated with at most one trait), making it hard to assess whether more than one trait is contributing evidence for association or which traits are associated with the variant.

The authors propose a summary-statistic-based likelihood ratio test of the omnibus null. This test aims to use a decomposition of the cross-trait correlation into "environmental" and genetic components to improve power. They then propose "m-values" (an adaptation of their earlier work on GWAS meta-analyses) to identify which traits contribute to a significant omnibus test (and in particular whether more than one trait contributes). The use of summary statistics (as opposed to individual level data) is an important practical advantage. The proposed method appears to outperform the approaches they've chosen for comparison in many situations (both simulated and empirical).

The paper could use one more close read for grammar and typos, but I commend the authors on their clarity and organization. (As somebody who dabbles in this space, I'm often overwhelmed deciding which comparisons, which simulation scenarios, which data are worth highlighting.)

My main concerns/suggestions are:

1) Expanding (slightly!) the set of comparison methods

2) Being more clear in the main text of the impact of sample overlap across traits on the relative performance of methods

As for #1: First, there are relatively simple multi-trait test statistics that both account for the correlation in test statistics due to sample overlap and define the rejection region differently (arguably more cleverly) than MIPAT (see PMIDs 28653391 and 31564761). It might be interesting to include some these in the power comparisons and in the visual illustration of the test rejection region (Fig 2), especially the "SUM" and "variance component" tests (PMID 28653391).

Also, the ASSET method (PMID 22560090) is widely used as a multivariate test of association and as an ad hoc method for detecting pleiotropy/identifying which traits are associated with a variant (those classified as "positively" or "negatively" associated with the effect allele). It would be quite interesting to compare the power/sensitivity/specificity of ASSET to the other methods in Tables 1 and 2.

As for #2: the potential gain for the authors’ proposed likelihood ratio test—which decomposes the correlation in test statistics into an “environmental” and genetic component—relative to other methods—which do not decompose the test statistic correlation (like MI GWAS or MPATs 28653391 or ASSET)—seems to depend on the magnitude of sample overlap. No or very little sample overlap=no or very little “environmental”contributions, and PAT collapses to something like a Wald test (28653391). This needs to be emphasized more strongly and earlier. The authors’ simulations and applications focus on scenarios where there is substantial overlap. Worth presenting simulations or theoretical results for relevant scenarios where there is less overlap (eg GWAS meta-analyses of schizophrenia and bipolar disease; breast and ovarian cancer; breast and prostate cancer [!])?

**Have all data underlying the figures and results presented in the manuscript been provided?**

Reviewer #1: None

Reviewer #2: None

Reviewer #3: Yes

PLOS authors have the option to publish the peer review history of their article (what does this mean?). If published, this will include your full peer review and any attached files.

Reviewer #1: No

Reviewer #2: No

Reviewer #3: No

---

## [Decision Letter · Decision Letter 1]

15 Aug 2022

Dear Dr Eskin,

Thank you very much for submitting your Research Article entitled 'Leveraging pleiotropy for joint analysis of genome-wide association studies with per trait interpretations' to PLOS Genetics. Reviewers 2 and 3 were satisfied with your responses to their previous concerns, whereas Reviewer 1 requested a bit more clarity on some matters. We therefore ask you to modify the manuscript to address Reviewer 1's comments (shown below). Once addressed, we expect to be  able to accept your Research Article promptly.

[LINK]

Yours sincerely,

Michael P. Epstein

Academic Editor

PLOS Genetics

David Balding

Section Editor

PLOS Genetics

Reviewer's **Comments to the Authors:**

Reviewer #1: I thank the authors for the detailed responses to my comments.

- As mentioned in my previous review, it is worth exploring the role of the direction of genetic correlation (positive/negative) on PAT's power: As you mentioned in the text, the results in Table 1 suggest that PAT outperforms other methods when height is included (it has a negative genetic correlation with other traits) but otherwise it underperforms when all traits happen to have positive genetic correlations. Does PAT always underperform compared to the other procedures whenever the genetic correlation structure between traits are in same direction (e.g. all positive), or is this an artifact of the selected traits? I think this is important point to explore and discuss in the paper.

Reviewer #2: The authors have addressed my comments.

Reviewer #3: The authors have done a nice job responding to the reviewers' comments. Really comprehensive, interesting stuff.

**Have all data underlying the figures and results presented in the manuscript been provided?**

Reviewer #1: Yes

Reviewer #2: None

Reviewer #3: Yes

PLOS authors have the option to publish the peer review history of their article (what does this mean?). If published, this will include your full peer review and any attached files.

Reviewer #1: No

Reviewer #2: No

Reviewer #3: No

---

## [Editor Report · Decision Letter 2]

27 Sep 2022

Dear Dr Eskin,

We are pleased to inform you that your manuscript entitled "Leveraging pleiotropy for joint analysis of genome-wide association studies with per trait interpretations" has been editorially accepted for publication in PLOS Genetics. Congratulations!

Yours sincerely,

Michael P. Epstein

Academic Editor

PLOS Genetics

David Balding

Section Editor

PLOS Genetics

Comments from the reviewers (if applicable):

**Data Deposition**

http://datadryad.org/submit?journalID=pgenetics&manu=PGENETICS-D-22-00245R2

**Press Queries**

---

## [Editor Report · Acceptance letter]

31 Oct 2022

PGENETICS-D-22-00245R2 

Leveraging pleiotropy for joint analysis of genome-wide association studies with per trait interpretations 

Dear Dr Eskin, 

We are pleased to inform you that your manuscript entitled "Leveraging pleiotropy for joint analysis of genome-wide association studies with per trait interpretations" has been formally accepted for publication in PLOS Genetics! Your manuscript is now with our production department and you will be notified of the publication date in due course.

With kind regards,

Zsuzsanna Gémesi

PLOS Genetics

On behalf of:
